# Continuous Two-Domain Equations of State for the Description of the Pressure-Specific Volume-Temperature Behavior of Polymers

**DOI:** 10.3390/polym12020409

**Published:** 2020-02-11

**Authors:** Jian Wang, Christian Hopmann, Malte Röbig, Tobias Hohlweck, Cemi Kahve, Jonathan Alms

**Affiliations:** 1College of Mechanical and Electrical Engineering, Beijing University of Chemical Technology, Beijing 100029, China; 2Institute for Plastics Processing (IKV), RWTH Aachen University, Aachen 52074, Germany; office@ikv.rwth-aachen.de (C.H.); malte.roebig@ikv.rwth-aachen.de (M.R.); tobias.hohlweck@ikv.rwth-aachen.de (T.H.); cemi.kahve@ikv.rwth-aachen.de (C.K.); Jonathan.Alms@ikv.rwth-aachen.de (J.A.)

**Keywords:** pvT, specific volume, polymer, equation of state, injection molding

## Abstract

The two-domain Schmidt equation of state (EoS), which describes the pressure-specific volume–temperature (pvT) behavior of polymers in both the equilibrium molten/liquid state and non-equilibrium solid/glassy state, is often used in the simulation of polymer processing. However, this empirical model has a discontinuity problem and low fitting accuracy. This work derived a continuous two-domain pvT model with higher fitting accuracy compared with the Schmidt model. The cooling rate as an obvious influencing factor on the pvT behavior of polymers was also considered in the model. The interaction parameters of the equations were fitted with the experimental pvT data of an amorphous polymer, acrylonitrile-butadiene-styrene (ABS), and a semi-crystalline polymer, polypropylene (PP). The fitted results by the continuous two-domain EoS were in good agreement with the experimental data. The average absolute percentage deviations were 0.1% and 0.16% for the amorphous and semi-crystalline polymers, respectively. As a result, the present work provided a simple and useful model for the prediction of the specific volume of polymers as a function of temperature, pressure, and cooling rate.

## 1. Introduction

The prediction of the pressure-specific volume-temperature (pvT) behavior of polymers is an important task, considering that it offers many highly promising applications in polymer physics and processing. Especially in injection molding, the predictions of shrinkage and warpage of the injection-molded parts are all based on pvT models [1,2,3,4]. In addition, the derived pvT diagrams have been proven to be very useful for optimal process control and mold design [5,6,7,8,9,10,11]. The pvT model can be used to map process variables to quality variables so that the course of cavity pressure can be adjusted to the actual path of melt temperature [6,7,8,11]. 

Polymers in molten/liquid state can be considered as equilibrium, thus the pvT behavior of the polymers in molten/liquid state can be represented accurately by the equation of state (EoS). Various theories have been developed to describe the pvT properties of polymer solutions, including the cell model [12,13,14], the hole model [15], the lattice-fluid model [16], and the statistical–mechanical EoS [17,18,19,20]. These theoretical models have been modified and extensively used [21,22]. However, the solid state is a quasi-equilibrium state, which is dependent on the conditions of pressure and cooling rate, and therefore is difficult to model by an analytical approach [21]. The semi-empirical model of Hartmann–Haque (HH) [23,24], on the other hand, showed satisfactory results for amorphous polymers but difficulties for semi-crystalline polymers [21]. Additionally, the specific volume is not a dependent variable in these theoretical and semi-empirical models, leading to difficulties in simulation or prediction. In comparison, empirical models are often convenient to represent pvT data on polymers in analytical form, by only giving the parameters in the correspondent equations [25]. Empirical pvT models include the Spencer–Gilmore model [26], the Schmidt model [27,28], and the Tait model [29,30], which are used in commercial software, e.g., by Moldflow (Autodesk, Inc.), Moldex 3D (CoreTech System Co., Ltd.), Sigmasoft (SIGMA Engineering GmbH) and Cadmould (Simcon kunststofftechnische Software GmbH). The Spencer–Gilmore model is quite simple but less accurate. The Tait model has been widely used for polymers [21,22,31,32], because of its simple form, convenient application, and high precision. The two-domain Tait EoS has much more fitting accuracy, especially in the solid state. The Schmidt model was developed by T.W. Schmidt at the Institute for Plastics Processing (IKV) in 1986 [28]. The two-domain Schmidt EoS is similar to the two-domain Tait EoS, although simpler, and it is primarily used in German-speaking countries [7,8,10]. The Tait model uses a volume–pressure relationship, while the Schmidt model considers a volume–temperature relationship. The Schmidt model has less accuracy than the Tait model [33], and fitting problems occur regularly. Furthermore, another important problem for the two-domain Tait EoS and the Schmidt EoS is the discontinuity of the models. Because the EoS describes the pvT behavior of the solid/glassy phase and molten/liquid phase separately, the two-domain equations consist of two independent sets of equations. The discontinuity of the specific volume occurs at the transition temperature, destabilizing the data prediction during the injection molding simulation [34]. The transition regime is crucial to estimate part shrinkage, since small changes in temperature and pressure lead to significant changes in the specific volume. Even a small discontinuity in the specific volume can cause large errors in the shrinkage estimates [35].

The cooling rate has an obvious influence on the pvT behavior of polymers [34,36,37,38,39,40]. In injection molding mainly non-isothermal conditions for polymer solidifications are found, with large cooling rates up to 3000 °C/min close to the mold walls and 60 °C/min near the center of the thickness [41]. Therefore, the prediction of pvT data of polymers considering the cooling rate is important. Chang et al. [42] presented a modified two-domain Tait EoS considering the cooling rate of the part regarding amorphous polymers. A pvT model for semi-crystalline polymers considering the cooling rate was also proposed [36], but the equations are inconvenient to use due to the increased experimental and analytical effort to fit the parameters in the crystallization range. Kowalska [43] summarized the correspondent criteria of the pvT EoS, e.g., that the EoS should be valid over a wide pressure and temperature range, not overly complex, and easy to use. 

In this paper, we derive a two-domain pvT model according to the Schmidt model. The new model, usable for both amorphous and semi-crystalline polymers, can solve the discontinuity problem of the traditional two-domain Schmidt and Tait EoS. Compared with the Schmidt model, the new model has higher accuracy of fitting; compared with the Tait model, the new model is simpler and easier to use. Moreover, the pvT behavior of polymers under different cooling rates can also be predicted. The interaction parameters of the equations are fitted with experimental pvT data. The prediction accuracy of the model is also validated. 

## 2. Materials and Methods 

### 2.1. Materials

Acrylonitrile-butadiene-styrene (ABS) (717C, CHIMEI Corporation, Zhenjiang, China) was used as the basic amorphous polymer material. The melt flow rate (MFR) of the ABS at 200 °C and 5 kg was 1.3 g/10min, the density was 1040 kg/m^3^ at room temperature of 23 °C. The semi-crystalline polymer used in the experiments was polypropylene (PP) (PP 505P, SABIC, Riyadh, Saudi Arabia). The melt flow rate (MFR) of the PP at 239 °C and 2.16 kg was 2.0 g/10min, the density was 905 kg/m^3^ at room temperature.

### 2.2. pvT Measurement

The pvT behavior of the two polymers was measured on a piston-die pvT instrument (type: PVT 500, GÖTTFERT Werkstoff-Prüfmaschinen GmbH, Buchen, Germany). Since the polymer in the mold cavity experiences mainly cooling (decreasing temperature) and decompression (decreasing pressure), the main pvT measurement mode of “isobaric cooling” and then “decompression” was performed. The initial isobaric pressure level was 2200 bar and was consecutively decreased to 1800, 1400, 1000, 600, and 200 bar for further isobaric measurement cycles. The temperature changed from 240 °C to 40 °C for the amorphous polymer ABS, and changed from 260 °C to 40 °C for the semi-crystalline polymer PP. Four cooling rates (20, 10, 5, and 2 °C/min) were used for the pvT device. Much higher cooling rates could not be realized due to the air-cooling system of the device. The exact cooling rates varied a lot for the set cooling rates of 10 and 20 °C/min. The data were recorded in temperature intervals of 5 °C. The pvT data of the isobaric pressure levels of 2200, 1800, 600, and 200 bar were used in the parameter regression of the equations, while the pvT data of the isobaric pressure levels of 1400 and 1000 bar were used to validate the prediction accuracy of the pvT model. 

### 2.3. Differential Scanning Calorimetry (DSC)

A differential scanning calorimeter (DSC-60, Shimadzu Corporation, Kyoto, Japan) was applied to validate the state transition temperature under atmospheric pressure. The temperature changed between 30 and 250 °C for the amorphous polymer ABS, and changed between 10 and 260 °C for the semi-crystalline polymer PP. Five heating and cooling rates (±50, ±30, ±10, ±5, and ±2 °C/min) were used. A holding time of 5 min was used after each heating and cooling process. 

### 2.4. Modeling

#### 2.4.1. Continuous Two-Domain EoS

Figure 1 presents the pvT curves for polymers. According to the pvT curves of amorphous polymers, linear expressions can be used to describe both the liquid state and glassy state. With semi-crystalline polymers, linear equations can be used for the molten state and a combination of linear and exponential expressions can be used for the solid state. Therefore, the equation to describe the pressure and temperature dependency of the specific volume, *v* (*p, T*), can be presented as:(1)v(p,T)=A(p)+B(p)·T+C(p,T)
where *A*(*p*) is the reference specific volume as a function of pressure, *B*(*p*) is the specific volume gradient to temperature, *C* (*p,T*) is the specific volume decrease due to crystallization. According to the Schmidt model [28], the pressure was taken into account by a hyperbolic approach for *A* (*p*) and *B* (*p*):(2)$$A\left(p\right)=\frac{a_{1}}{a_{2}+p}$$
(3)$$B\left(p\right)=\frac{b_{1}}{b_{2}+p}$$

An exponential function is usually used in the Tait model, however, it was found that a polynomial function fits better [44]. In order to improve the fitting accuracy, a quadratic polynomial expression was used to replace the hyperbolic expressions in the Schmidt model: (4)$$A\left(p\right)=a_{1}-a_{2}\cdot p+a_{3}\cdot p^{2}$$
(5)$$B\left(p\right)=b_{1}-b_{2}\cdot p+b_{3}\cdot p^{2}$$

Constants *a*_1_ to *a*_3_ are parameters that describe the dependency of the reference specific volume on pressure. Constants *b*_1_ to *b*_3_ are the parameters to describe the dependence of the specific volume gradient on pressure. A comparison between the polynomial model and the Schmidt model is discussed in this work. The description of the pvT behavior in the crystallization range is approximated by the following exponential function: (6)$$C\left(p,T\right)=c_{1}\cdot exp\left(c_{2}\cdot T-c_{3}\cdot p\right)$$
where *c*_1_, *c*_2_ and *c*_3_ are particular parameters of semi-crystalline polymers that describe the form of the crystallization state. For amorphous polymers, *C* (*p, T*) equals zero. The state transition temperature increases with the pressure. The traditional Tait EoS and Schmidt EoS give the linear rule for the effect of pressure on the transition temperature.
(7)$$T_{t}\left(p\right)=d_{1}+d_{2}\cdot p$$

However, many published articles [45,46] have shown a nonlinear relationship between the transition temperature and the pressure. Therefore, we used a quadratic polynomial Equation in order to have a more accurate model: (8)$$T_{t}\left(p\right)=d_{1}+d_{2}\cdot p+d_{3}\cdot p^{2}$$
where *d*_1_, *d*_2_, and *d*_3_ are the constants that describe the transition temperature as a function of pressure. 

The two domains are related to two states of polymers, solid state and molten state for semi-crystalline polymer, glass state and liquid state for amorphous polymer. The transition temperature (*T_t_*) is the crystallization temperature (*T_c_*) for semi-crystalline polymer and the glass transition temperature (*T_g_*) for amorphous polymer. To deal with the discontinuity problem of the traditional two-domain EoS, the two-domain equations should be connected by the transition temperature and the corresponding specific volume. Thus, we used a reference temperature, $\bar{T}$, instead of T in the Equations:(9)$$\bar{T}\left(p,T\right)=T-T_{t}\left(p\right)$$
where *T* is the exact temperature. It was noted that the equation, $\bar{T}=T-d_{1}$, has been used in the two-domain Tait EoS in many publications [11,22,30,42]. However, it will cause large deviations without the dependency on pressure [34], because *d*_1_ only represents the transition temperature at zero pressure, which cannot combine the pressure effects. Equation (9) combined with Equation (8) becomes:(10)$$\bar{T}\left(p,T\right)=T-d_{1}-d_{2}\cdot p-d_{3}\cdot p^{2}$$

The material-dependent constants *A*, *B*, and *C* differ for the two domains. Therefore, the following sets of equations can be used for specific cases:

When $T>T_{t}\left(p\right)$
(11)$$v\left(p,T\right)=A_{m}+B_{m}\cdot \bar{T}$$
(12)$$A_{m}=\frac{a_{1m}}{a_{2m}+p}$$
or:(13)$$A_{m}=a_{1m}-a_{2m}\cdot p+a_{3m}\cdot p^{2}$$
(14)$$B_{m}=\frac{b_{1m}}{b_{2m}+p}$$
or:(15)$$B_{m}=b_{1m}-b_{2m}\cdot p+b_{3m}\cdot p^{2}$$
when $T<T_{t}\left(p\right)$:(16)$$v\left(p,T\right)=A_{s}+B_{s}\cdot \bar{T}+C$$
(17)$$A_{s}=\frac{a_{1s}}{a_{2s}+p}$$
or:(18)$$A_{s}=a_{1s}-a_{2s}\cdot p+a_{3s}\cdot p^{2}$$
(19)$$B_{s}=\frac{b_{1s}}{b_{2s}+p}$$
or:(20)$$B_{s}=b_{1s}-b_{2s}\cdot p+b_{3s}\cdot p^{2}$$
(21)$$C=c_{1}\cdot exp\left(c_{2}\cdot \bar{T}-c_{3}\cdot p\right)$$

The subscripts “*m*” and “*s*” denote the parameter values for the molten/liquid and solid/glassy states, respectively. The estimation of these constants is conducted separately. Figure 2a shows the order of the parameter regression for the discontinuous two-domain EoS. Firstly, *d*_1_, *d*_2_, and *d*_3_ can be estimated by Equation (8) from the data of transition temperature at different pressures, and then the parameter values are used in the regression of the molten/liquid and glassy/solid parameters. The molten/liquid parameters (*a*_1m_, *a*_2m_, *a*_3m_, *b*_1m_, *b*_2m_, and *b*_3m_) and glassy/solid parameters (*a*_1s_, *a*_2s_, *a*_3s_, *b*_1s_, *b*_2s_, *b*_3s_, *c*_1_, *c*_2_, and *c*_3_) can be estimated separately from the corresponding data. *C* (*c*_1_, *c*_2_, and *c*_3_) equals zero for amorphous polymers.

In order to connect the two domains, the key parameters are the transition temperature, *T_t_*, and the correlating specific volume, *v_t_*. When the temperature equals *T_t_*, the specific volume of the solid/glassy state, *v*_s_, and the specific volume of the molten/liquid state, *v*_m_, should be equal. For this case, $T=T_{t}$, $\bar{T}=T-T_{t}=0$, the specific volume at the transition temperature is assigned as:(22)$$v_{m}=A_{m}=v_{s}=A_{s}$$
for amorphous polymers and:(23)$$v_{m}=A_{m}=v_{s}=A_{s}+c_{1}\cdot exp\left(-c_{3}\cdot p\right)$$
for semi-crystalline polymers. The coefficients equal to:(24)$$c_{1}\cdot exp\left(-c_{3}\cdot p\right)=A_{m}-A_{s}$$
(25)$$A_{s}=A_{m}-c_{1}\cdot exp\left(-c_{3}\cdot p\right)$$

Therefore, the continuous two-domain equations with considering the transition points could be derived as Equations (8) and (9).

When $T>T_{t}\left(p\right)$, see Equation (11), when $T<T_{t}\left(p\right)$,
(26)$$v\left(p,T\right)=A_{m}\left(p\right)+B_{s}\left(p\right)\cdot \bar{T}$$
for amorphous polymers,
(27)$$v\left(p,T\right)=A_{s}\left(p\right)+B_{s}\left(p\right)\cdot \bar{T}+\left[A_{m}\left(p\right)-A_{s}\left(p\right)\right]\cdot exp\left(c_{2}\cdot \bar{T}\right)$$
or,
(28)$$v\left(p,T\right)=A_{m}\left(p\right)+B_{s}\left(p\right)\cdot \bar{T}+c_{1}\;exp\left(-c_{3}\cdot p\right)\;\left[exp\left(c_{2}\;\bar{T}\right)-1\right]$$
for semi-crystalline polymers.

The modified Equations (26)–(28) established a link between the two states of polymers. The specific volume equals *A_m_*(*p*) when $\bar{T}$ equals zero—*A_m_*(*p*) is the specific volume at the transition temperature. When the transition temperature and the correspondent specific volume have been fixed, the parameters for the two domains can be fitted separately. The new two-domain EoS does not only solve the problem of discontinuity, but also saves two or more parameters. For amorphous polymers, *A_m_*(*p*) is used directly to replace *A_s_*(*p*); for semi-crystalline polymers, *c*_1_ and *c*_3_, can be saved by Equation (27), or *A_s_*(*p*) can be saved by Equation (28). Equation (28) is preferred over Equation (27) because it can save more parameters. Finally, Equations (8), (9), (11), (26), and (28) in combination with Equations (12–15), (19) and (20) form the continuous two-domain pvT model. Fifteen parameters are included in these equations. Figure 2b presents the order of the parameter regression for the continuous two-domain EoS. The values of *d*_1_, *d*_2_, and *d*_3_ are firstly estimated by Equation (8) from the data of transition temperature at different pressures, and the values of *a*_1m_=*a*_1s_=*a*_1_, *a*_2m_=*a*_2s_=*a*_2_, and *a*_3m_=*a*_3s_=*a*_3_ are estimated by Equation (13) or (18) from the data of transition specific volume at different pressures. Then, the estimated values of *d*_1_, *d*_2_, *d*_3_, *a*_1_, *a*_2_, and *a*_3_ are used in the following regressions. The melt/liquid parameters (*b*_1m_, *b*_2m_, and *b*_3m_) and solid/glassy parameters (*b*_1s_, *b*_2s_, *b*_3s_, *c*_1_, *c*_2_, and *c*_3_) can be estimated separately. 

#### 2.4.2. Two-Domain EoS Considering Cooling Rate

In principle, the proposed functions are furthermore able to determine the cooling rate dependent pvT behavior. Dependencies between the transition temperature and the specific volume need to be determined regarding the cooling rate. The parameters on the transition temperature and their specific volume with cooling rate effects are *d*_1_ and *a*_1_. Chang et al. [42] suggested a logarithmic relationship between the transition temperature and the cooling rate. When the cooling rate increases from ${\dot{T}}_{0}$ to $\dot{T}=dT/dt$, the transition temperature shifts and the corresponding parameter *d*_1_ can be integrated.
(29)$$d_{1}=d_{1}^{\prime }+d_{1}^{\prime \prime }\cdot \ln\left(q\right)$$
(30)$$q=\frac{\dot{T}}{{\dot{T}}_{0}}$$

Constants *d*_1′_ and *d*_1′_*’* are two parameters to describe the transition temperature as a function of the cooling rate. The correspondent transition specific volume also changes to:(31)$$a_{1}=a_{1}^{\prime }+a_{1}^{\prime \prime }\cdot \ln\left(q\right)$$
where the constants *a*_1′_ and *a*_1′_*’* are two parameters to describe the specific volume at the transition temperature as a function of the cooling rate.

The continuous two-domain equations including cooling rate effects to describe the pvT behavior of both amorphous and semi-crystalline polymers are presented in Equations (29)–(31) and following final set of equations: (32)$$v\left(p,T,q\right)=A\left(p,q\right)+B\left(p\right)\cdot \bar{T}\left(p,T,q\right)+C\left(p,T,q\right)$$
(33)$$A\left(p,q\right)=a_{1}-a_{2}\cdot p+a_{3}\cdot p^{2}$$
(34)$$\bar{T}\left(p,T,q\right)=T-T_{t}\left(p,q\right)$$
(35)$$T_{t}\left(p,q\right)=d_{1}+d_{2}\cdot p+d_{3}\cdot p^{2}$$
when $T>T_{t}\left(p,\;q\right)$, see Equation (15) and
(36)C(p,T,q)=0
when $T<T_{t}\left(p,q\right)$, see Equation (20) and
(37)$$C\left(p,T,q\right)=c_{1}\cdot exp\left(-c_{3}\cdot p\right)\cdot \left[exp\left(c_{2}\cdot \bar{T}\right)-1\right]$$

Seventeen parameters are included in these equations. Figure 3 presents the modeling parameters. The values of *d*_1′_, *d*_1′_’, *d*_2_, and *d*_3_ are estimated by Equation (35) and Equation (29) from the data of transition temperature at different pressures and cooling rates. The values of *a*_1′_, *a*_1′_’, *a*_2_, and *a*_3_ are estimated by Equation (33) and Equation (31) from data of the specific volume at the transition temperature. For amorphous polymers, *c*_1_, *c*_2_, and *c*_3_ all equal zero. For the cooling rate, 0.001 °C/s is used as the initial value of the cooling rate, ${\dot{T}}_{0}$, due to the asymptote of the natural logarithm at zero. 

## 3. Results and Discussion

### 3.1. Influence of the Ccooling Rate

The effect of the cooling rate cannot be neglected regarding the pvT behavior of polymers [36,37,42]. However, the cooling rate is not constant during the pvT measurement as well as in the injection molding process. Figure 4 shows the cooling rate as a function of the temperature during the pvT measurement. The cooling rate was calculated from the gradient of the temperature over time. Cooling rates of 2 and 5 °C/min were obtained accurately, but cooling rates of 10 and 20 °C/min were not realized well. The maximum cooling rates for the set cooling rates of 10 and 20 °C/min were around 9.6 and 12.4 °C/min, respectively. The cooling rate varied a lot during the starting and ending phases of the measurement cycle. In reality, the cooling rate is never constant because it dependents on the heat flux and the temperature of the coolant in combination with the heat capacity of the sample-holding area. Therefore, the actual temperature history was used in the analysis of the experimental results, where the maximum in the time derivative of the temperature was defined as the cooling rate $\dot{T}$ [36], which was used for the fitting of the parameters. It should be noted that the maximum cooling rate in injection molding can be as high as 3000 °C/min [41]. Several laboratories have reported that they can realize equally high cooling rates [36,41,45], but no commercial pvT instrument is available with such abilities. This further indicates the necessity of predictive models to extrapolate pvT data for higher cooling rates. Due to the limitation of the experimental device, only cooling rates up to 12.4 °C/min were used in this study. 

### 3.2. Transition Temperature and the Corresponding Specific Volume

Regarding the regression of the two-domain equations, the pvT data at the transition phase are very important. In order to confirm the accuracy, for each group of isobaric pvT data, we used linear equations to fit the data for the molten state and quadratic polynomial equations to fit the data for the crystallization state. Then, we calculated the data at the transition phase, which is the intersect point of these two equations. Data of the transition temperature and pressure were used in the regression of Equations (35) and (29), and the data of the corresponding specific volume and pressure were used in the regression of Equations (33) and (31). Figure 5 shows the transition temperature, *T*_t_, as a function of the cooling rate. The glass transition temperature, *T*_g_, of the amorphous polymer increased with increasing cooling rate, and the transition/crystallization temperature, *T*_c_, of the semi-crystalline polymer decreased. The data were fitted very well. The estimated parameter values are shown in Table 1. The R-squared (*R*^2^) value from the quadratic polynomial regression model was 99.7% for both amorphous and semi-crystalline polymers. The predictions and additional experimental results at isobaric pressure levels of 1400 and 1000 bar were mostly consistent. In addition, DSC was used to obtain the state transition temperatures of the ABS and PP samples under atmospheric pressure. The comparison of state transition temperatures between the DSC and the prediction is shown in Figure 5. Good agreement could be obtained especially for the semi-crystalline PP. Due to the process dependency, the amorphous polymer did not exhibit clear glass transition temperatures. 

Figure 6 shows the corresponding specific volume at the transition temperature, *v_t_*, as a function of the cooling rate. As the cooling rate increased, the corresponding specific volume of the amorphous ABS increased. The corresponding specific volume of the semi-crystalline PP increased but less than the amorphous polymer. The calculated data by Equations (33) and (31) are shown by “vt_cal.1”, with good matching of experimental data and predictions. For comparison, Equation (2) from the Schmidt model combined with Equation (31) was also used in the fitting. However, the predicted results as shown by “vt_cal.2” present a larger deviation than the quadratic polynomial equation. Table 2 lists different parameters for the corresponding specific volume at the transition temperature. The polynomial Equation (33) combined with Equation (31) was compared with the Schmidt Equation (2) combined with Equation (31). From the R-squared (*R*^2^) values, the Schmidt model accounted for 99.7% of the variance, while the polynomial model accounted for 99.9%. Therefore, the polynomial approach seems to have less error for both materials, ABS and PP.

### 3.3. Comparison between the Experimental Data and the Fitted Data

Figure 7 shows the comparison between the experimental data and the fitted data by discontinuous and continuous pvT models. The “m” and “s” denote the parameter values for the molten/liquid and solid/glassy states, respectively. The quadratic polynomial model and Schmidt model were used separately in the description of the pressure dependence, and their fitted results are shown by “cal. 1” and “cal. 2” respectively. Good agreement between the experimental and the calculated specific volumes was obtained. In comparison with the Schmidt model (cal. 2), smaller deviation and better fitting can be realized by the quadratic polynomial model (cal. 1). No significant difference between the discontinuous and continuous models can be seen in the fitting of experimental results. The estimated values of these characteristic parameters are shown in Table 3 and Table 4. The *R*^2^ values of the quadratic polynomial model (cal. 1) were all higher than those of the Schmidt model (cal. 2). In comparison with the discontinuous model, the *R*^2^ values of the continuous model were slightly lower due to the limiting situation of the specific volume at the transition temperature. 

To quantify the accuracies of the models, the average absolute percentage deviation, $PD_{ave}$, was calculated by the following Equations:(38)$$PD_{i}=100\times \frac{\left(v_{i,exp}-v_{i,cal}\right)}{v_{i,exp}}$$
(39)$$PD_{ave}=\frac{\sum_{i=1}^{n}\left|PD_{i}\right|}{n}$$
where $v_{i,exp}$ is the specific volume, $v_{i,cal}$ is the calculated specific volume, and $PD_{i}$ is the percentage deviation. Figure 8 shows the average absolute percentage deviation for all the fitting with different models. The polynomial model had a lower deviation than the Schmidt model. The continuous model had a slightly higher deviation than the discontinuous model. The average absolute percentage deviations of the fitting by the continuous polynomial model for the amorphous ABS and semi-crystalline PP were 0.1% and 0.16%, respectively.

### 3.4. Validation of the Models

Figure 9 shows the comparison between the experimental results and the prediction results at two isobaric levels of 1400 and 1000 bar. Experimental data at the two isobaric pressures (1400, 1000 bar) were not used in the regression, they were only used for validation. Good prediction was shown by both of the continuous and discontinuous models. The *R*^2^ values in Figure 10 show better prediction accuracy by the continuous model in the molten/liquid state but worse accuracy in the solid state of the semi-crystalline PP compared with the discontinuous model. Figure 11 shows the average absolute percentage deviations for all the validation with different models. In comparison with the Schmidt model, the polynomial model possessed higher ability to predict with lower error. The continuous model, although it provided a good correlation in prediction, had a slightly lower accuracy than the discontinuous model. The average absolute percentage deviations of the continuous polynomial model were 0.09% and 0.17% for the amorphous ABS and semi-crystalline PP, respectively.

### 3.5. Prediction with the Model 

The superiority of the continuous EoS became significant when the models with their estimated parameters were used in the prediction. Figure 12 shows the prediction data at the pressures of 0, 1500, and 2500 bar and the cooling rates of 0 and 1800 °C/s. The discontinuous problems are shown clearly in the prediction results by the discontinuous models. Large deviations in the specific volume at the transition phase could be seen using the discontinuous model. The deviations at the transition phase were much larger for the semi-crystalline polymer than for the amorphous polymer. The continuous equations performed very well in the prediction, even at high pressures and high cooling rates. It was noted that the pvT data at the pressure of 0 bar and the cooling rates of 0, 1800 °C/s could not be measured by the experimental device. Due to the limitation of the pvT measurement device for much faster cooling, the four maximum cooling rates of 2, 5, 9.6, and 12.4 °C/min used in this work may not be enough to confirm the accuracy of Equation (31). For much higher cooling rates, it still needs more experimental data to validate the model and then the related parameters of the model, such as *a*_1′_, *a*_1′_*’*, *a*_2_, and *a*_3_, could be modified. 

## 4. Conclusions

In order to deal with the discontinuity problems of the traditional two-domain EoS in the description of the pvT behavior of polymers and further improve the accuracy of the models, a continuous two-domain pvT model was derived in this work. The predicted results by this proposed pvT model can closely correlate with the experimental data, both performed in amorphous polymers and semi-crystalline polymers. The quadratic polynomial equations were used in the modeling of the pressure dependence regarding the transition temperature, the specific volume, and the specific volume gradient. In comparison with the Schmidt model, the polynomial model showed higher accuracy. The two-domain pvT data for the two states of polymers (molten and solid states for semi-crystalline polymers, and liquid and glassy states for amorphous polymers) were connected by the fixed transition temperatures and their correspondent specific volumes. Although the accuracy of the continuous model was slightly lower than the discontinuous model due to the limiting accuracy of the specific volume at the transition temperature, no deviation of specific volume at the transition phase occurred when using the continuous model in prediction. The new pvT model can also be applied to describe the influence of the cooling rate. Logarithmic functions were used to model the cooling rate dependence regarding the transition temperature and the corresponding specific volume. High R-squared values, >99.43%, and low average absolute percent deviations, <0.16%, in the regression of the continuous two-domain EoS were obtained. The results presented here could be employed to improve the prediction accuracy of the shrinkage and warpage of polymer processing. 

## Figures and Tables

**Figure 1 polymers-12-00409-f001:**
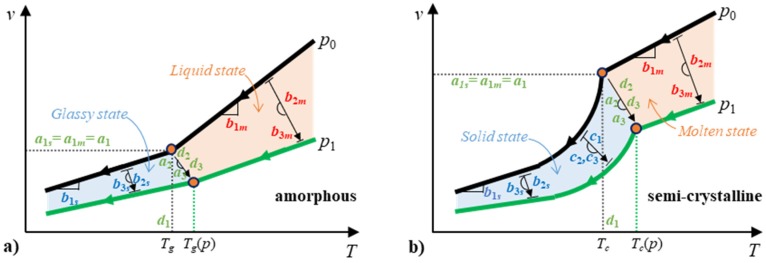
Schematic of the pressure-specific volume–temperature (pvT) curves for amorphous (**a**) and semi-crystalline (**b**) polymers and the relative parameters for the continuous two-domain equation of state (EoS).

**Figure 2 polymers-12-00409-f002:**
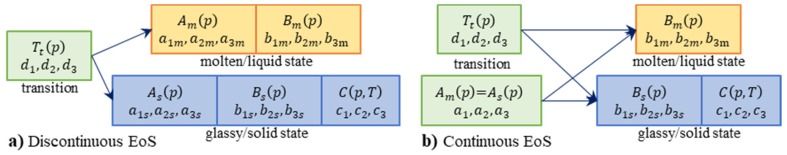
Parameter regression order of the discontinuous (**a**) and continuous (**b**) two-domain EoS.

**Figure 3 polymers-12-00409-f003:**
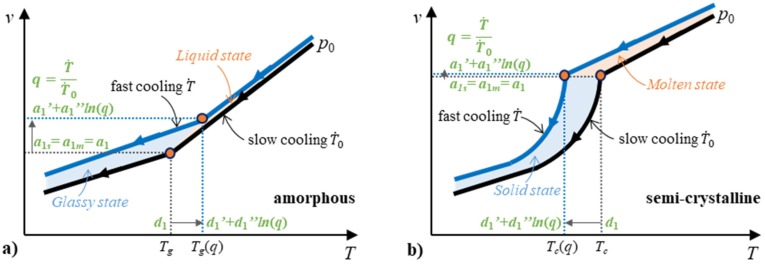
Schematic of the cooling rate-dependent pvT curves for amorphous (**a**) and semi-crystalline (**b**) polymers and the relative parameters for the continuous two-domain EoS considering the cooling rate.

**Figure 4 polymers-12-00409-f004:**
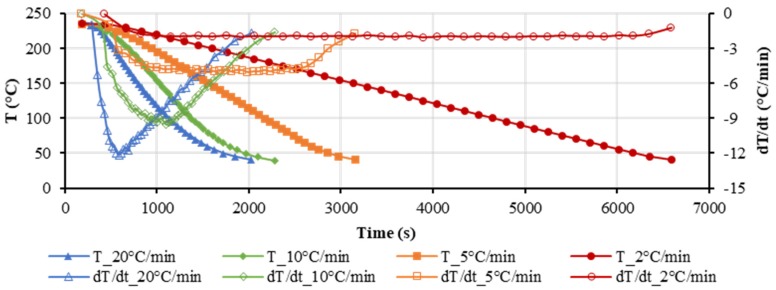
Actual temperature and cooling rate as a function of time during the pvT measurement.

**Figure 5 polymers-12-00409-f005:**
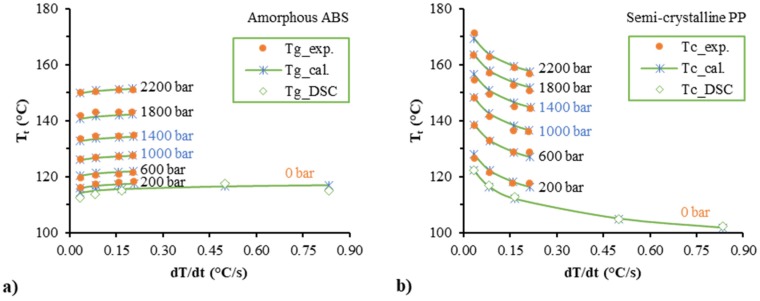
Experimental and calculated transition temperature of the amorphous Acrylonitrile-butadiene-styrene (ABS) (**a**) and the semi-crystalline polypropylene (PP) (**b**) as a function of the cooling rate.

**Figure 6 polymers-12-00409-f006:**
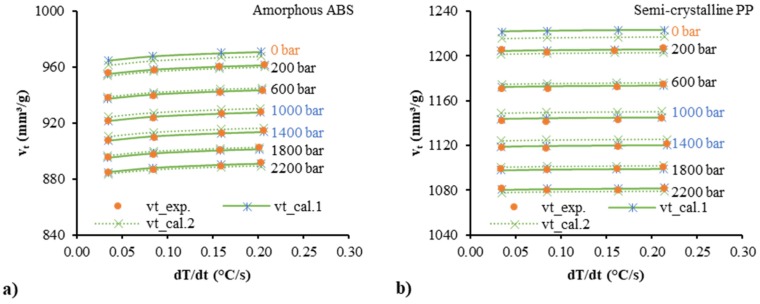
The corresponding specific volume at the transition temperature of the amorphous ABS (**a**) and the semi-crystalline PP (**b**) as a function of the cooling rate.

**Figure 7 polymers-12-00409-f007:**
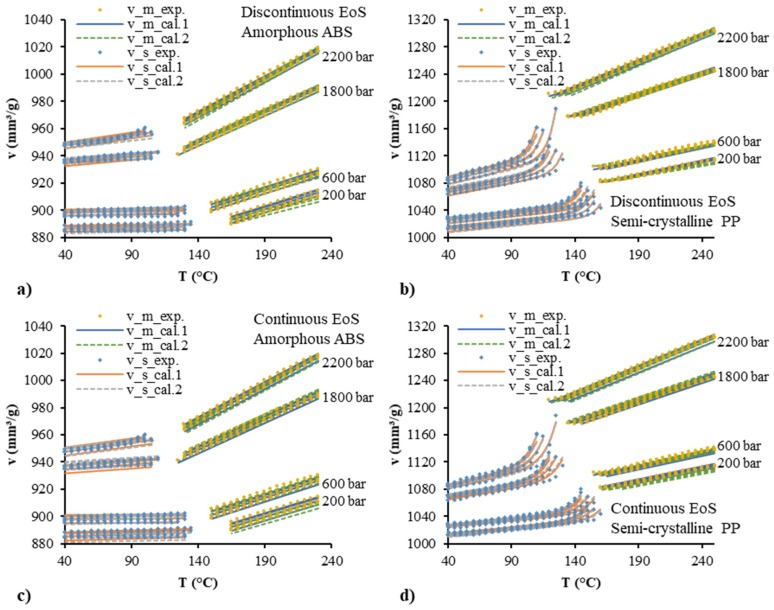
Comparison between the experimental pvT data and the calculated data by the discontinuous two-domain EoS for amorphous ABS (**a**) and semi-crystalline PP (**b**) and the continuous two-domain EoS for amorphous ABS (**c**) and semi-crystalline PP (**d**).

**Figure 8 polymers-12-00409-f008:**
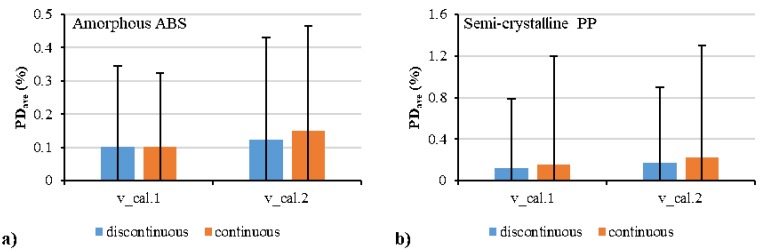
Average absolute percentage deviations between the calculated data from the experimental data: amorphous polymer (**a**) and semi-crystalline polymer (**b**).

**Figure 9 polymers-12-00409-f009:**
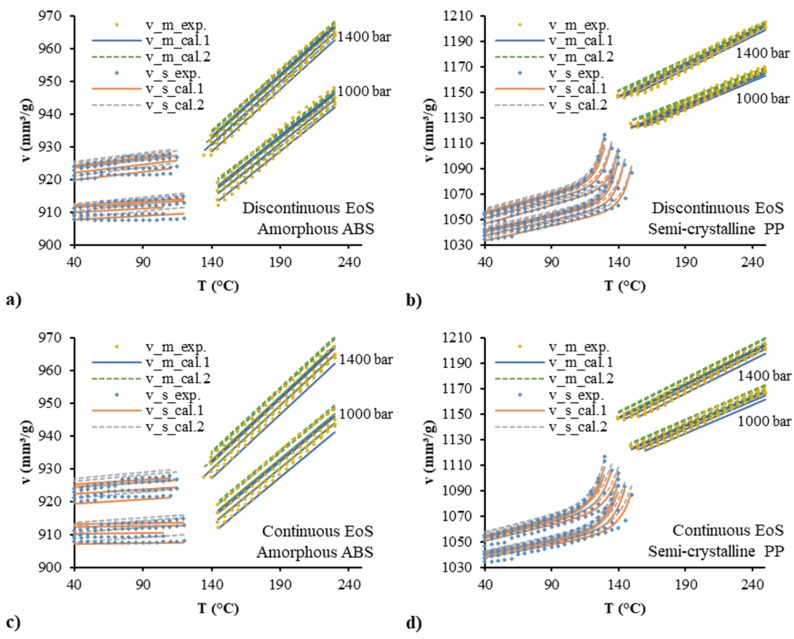
Validation of the discontinuous two-domain EoS for amorphous ABS (**a**) and semi-crystalline PP (**b**) and the continuous two-domain EoS for amorphous ABS (**c**) and semi-crystalline PP (**d**) with the additional pvT data at two isobaric levels of 1400 and 1000 bar.

**Figure 10 polymers-12-00409-f010:**
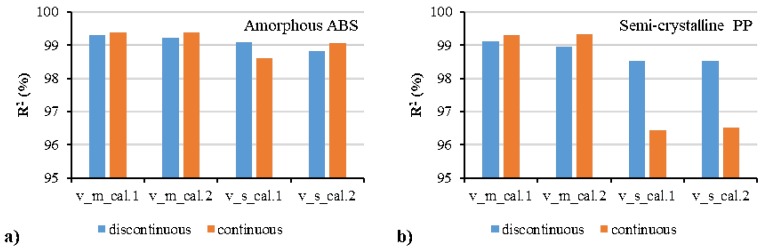
R-squared values of different models for the validation with the additional pvT data at two isobaric levels of 1400 and 1000 bar: amorphous ABS (**a**) and semi-crystalline PP (**b**).

**Figure 11 polymers-12-00409-f011:**
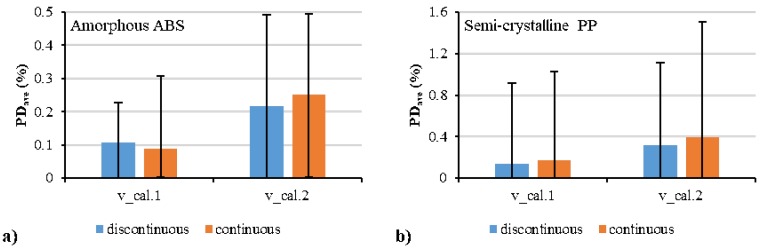
Average absolute percentage deviations between the specific volume calculated by different models and the additional experimental specific volume at two isobaric levels of 1400 and 1000 bar: amorphous ABS (**a**) and semi-crystalline PP (**b**).

**Figure 12 polymers-12-00409-f012:**
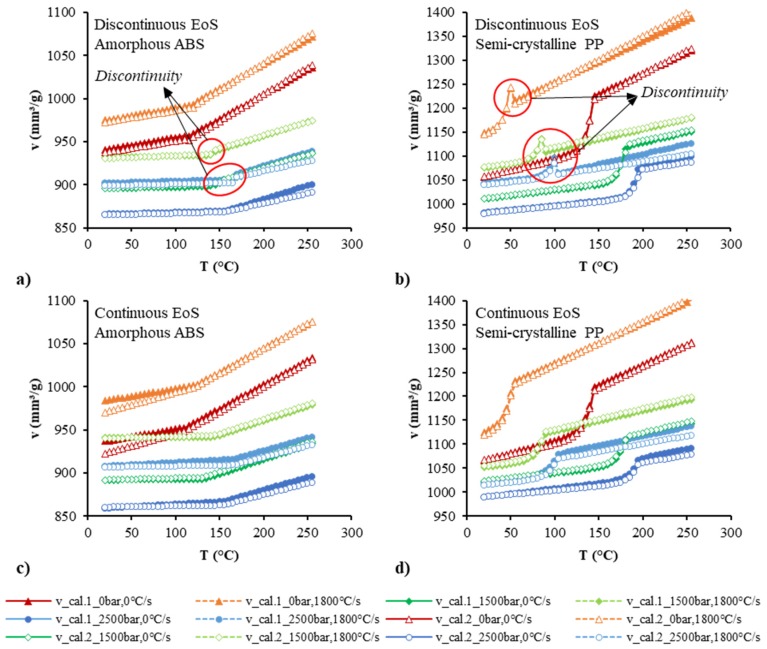
Predicted specific volume as a function of temperature, pressure, and cooling rate by the discontinuous two-domain EoS for amorphous polymers (**a**) and semi-crystalline polymers (**b**) and the continuous two-domain EoS for amorphous polymers (**c**) and semi-crystalline polymers (**d**).

**Table 1 polymers-12-00409-t001:** Parameters for the transition temperature, *T*_t_, estimated by the quadratic polynomial regression model considering the cooling rate, Equations (35) and (29).

Polymer	Parameter	Estimate	Std. Error	*R*^2^ (%)
ABS	d_1_’ (°C)	111.217	1.536	99.7
	d_1_’’ (°C)	0.865	0.306	
	d_2_ (°C/bar)	0.008	0.002	
	d_3_ (°C/bar^2^)	3.74 × 10^−6^	6.58 × 10^−7^	
PP	d_1_’ (°C)	144.71	2.053	99.7
	d_1_’’ (°C)	−6.401	0.406	
	d_2_ (°C/bar)	0.03	0.002	
	d_3_ (°C/bar^2^)	3.852 × 10^−6^	8.94 × 10^−7^	

**Table 2 polymers-12-00409-t002:** Parameters for the correspondent specific volume at the transition temperature, vt, estimated by the continuous two-domain EoS considering the cooling rate.

Polymer	Model	Equation	Parameter	Estimate	Std. Error	*R*^2^ (%)
ABS	cal.1	(33)	a_1_’ (mm^3^/g)	952.781	1.601	99.9
	Polynomial	(31)	a_1_’’ (mm^3^/g)	3.414	0.319	
			a_2_ (mm^3^/(g·bar))	0.049	0.002	
			a_3_ (mm^3^/(g·bar^2^))	5.789 × 10^−6^	6.86 × 10^−7^	
	cal.2	(2)	a_1_’ (mm^3^·bar/g)	23,649,719	374,659	99.7
	Schmidt	(31)	a_1_’’ (mm^3^·bar/g)	88,851	17,092	
			a_2_ (bar)	24,925.451	403.474	
PP	cal.1	(33)	a_1_’ (mm^3^/g)	1219.121	2.534	99.9
	Polynomial	(31)	a_1_’’ (mm^3^/g)	0.704	0.502	
			a_2_ (mm^3^/(g·bar))	0.089	0.003	
			a_3_ (mm^3^/(g·bar^2^))	1.1483 × 10^−5^	1.103 × 10^−6^	
	cal.2	(2)	a_1_’ (mm^3^·bar/g)	20,850,700	339,448	99.7
	Schmidt	(31)	a_1_’’ (mm^3^·bar/g)	13,276	20,676	
			a_2_ (bar)	17,190.134	285.183	

**Table 3 polymers-12-00409-t003:** Parameters estimated by the continuous two-domain EoS considering the cooling rate.

Polymer	Model	Equation	Parameter	Estimate	Std. Error	*R*^2^ (%)
ABS	cal.1	(32)	b_1m_ (mm^3^/(g·°C))	0.559	0.003	99.91
		(15)	b_2m_ (mm^3^/(g·°C·bar))	2.11521 × 10^−4^	8.131 × 10^−6^	
			b_3m_ (mm^3^/(g·°C·bar^2^))	4.2 × 10^−8^	3 × 10^−9^	
		(32)	b_1s_ (mm^3^/(g·°C))	0.175	0.004	99.81
		(20)	b_2s_ (mm^3^/(g·°C·bar))	2.1499 × 10^−4^	9.644 × 10^−6^	
			b_3s_ (mm^3^/(g·°C·bar^2^))	6.7 × 10^−8^	4 × 10^−9^	
	cal.2	(32)	b_1m_ (mm^3^/(g·°C))	1216.716	29.363	99.79
		(14)	b_2m_ (bar)	2113.472	61.924	
		(32)	b_1s_ (mm^3^/(g·°C))	44.271	3.652	99.72
		(19)	b_2s_ (bar)	154.237	33.575	
PP	cal.1	(32)	b_1m_ (mm^3^/(g·°C))	0.846	0.005	99.90
		(15)	b_2m_ (mm^3^/(g·°C·bar))	4.14538 × 10^−4^	1.3043 × 10^−5^	
			b_3m_ (mm^3^/(g·°C·bar^2^))	9.2 × 10^−8^	6 × 10^−9^	
		(32)	b_1s_ (mm^3^/(g·°C))	0.496	0.023	99.43
		(20)	b_2s_ (mm^3^/(g·°C·bar))	3.31605 × 10^−4^	2.6266 × 10^−5^	
		(37)	b_3s_ (mm^3^/(g·°C·bar^2^))	8.8 × 10^−8^	8 × 10^−9^	
			c_1_ (mm^3^/g)	90.531	1.336	
			c_2_ (1/°C)	0.109	0.003	
			c_3_ (1/bar)	3.1409 × 10^−4^	8.201 × 10^−6^	
	cal.2	(32)	b_1m_ (mm^3^/(g·°C))	1280.478	29.722	99.79
		(14)	b_2m_ (bar)	1432.768	41.907	
		(32)	b_1s_ (mm^3^/(g·°C))	713.539	73.594	99.19
		(19)	b_2s_ (bar)	1718.018	278.007	
		(37)	c_1_ (mm^3^/g)	94.12	1.654	
			c_2_ (1/°C)	0.104	0.003	
			c_3_ (1/bar)	3.4312 × 10^−4^	9.75 × 10^−6^	

**Table 4 polymers-12-00409-t004:** Parameters estimated by the discontinuous two-domain EoS considering the cooling rate.

Polymer	Model	Equation	Parameter	Estimate	Std. Error	*R*^2^ (%)
ABS	cal.1	(11)	a_1m_’ (mm^3^/g)	954.108	0.598	99.91
		(13)	a_1m_’’ (mm^3^/g)	2.945	0.093	
		(15)	a_2m_ (mm^3^/(g·bar))	0.046	0.001	
		(31)	a_3m_ (mm^3^/(g·bar^2^))	4.668 × 10^−6^	4.49 × 10^−7^	
			b_1m_ (mm^3^/(g·°C))	0.571	0.006	
			b_2m_ (mm^3^/(g·°C·bar))	2.49678 × 10^−4^	1.7266 × 10^−5^	
			b_3m_ (mm^3^/(g·°C·bar^2^))	5.9 × 10^−8^	7 × 10^−9^	
		(16)	a_1s_ (mm^3^/g)	956.846	0.603	99.81
		(18)	a_2s_ (mm^3^/(g·bar))	2.535	0.088	
		(20)	a_3s_ (mm^3^/(g·bar^2^))	4.6 × 10^−8^	1 × 10^−9^	
		(31)	b_1s_ (mm^3^/(g·°C))	4.344	0.448	
			b_2s_ (mm^3^/(g·°C·bar))	1.79 × 10^−7^	9 × 10^−9^	
			b_3s_ (mm^3^/(g·°C·bar^2^))	1.75542 × 10^−4^	1.89 × 10^−5^	
	cal.2	(11)	a_1m_’ (mm^3^·bar/g)	25,081,194	30,453	99.79
		(12)	a_1m_’’ (mm^3^·bar/g)	80,827	60,364	
		(14)	a_2m_ (bar)	26,415.869	3.761	
		(31)	b_1m_ (mm^3^·bar /(g·°C))	906.314	170.66	
			b_2m_ (bar)	1452.427	156.634	
		(16)	a_1s_’ (mm^3^·bar/g)	24,400,643	111,151	99.72
		(17)	a_1s_’’ (mm^3^·bar/g)	66,735	3007	
		(19)	a_2s_ (bar)	25,614.391	121.87	
		(31)	b_1s_ (mm^3^·bar /(g·°C))	75.047	8.844	
			b_2s_ (bar)	446.842	81.5	
PP	cal.1	(11)	a_1m_’ (mm^3^/g)	1225.188	0.87	99.9
		(13)	a_1m_’’ (mm^3^/g)	-0.789	0.143	
		(15)	a_2m_ (mm^3^/(g·bar))	0.086	0.001	
		(31)	a_3m_ (mm^3^/(g·bar^2^))	1.0111 × 10^−5^	6.19 × 10^−7^	
			b_1m_ (mm^3^/(g·°C))	0.857	0.008	
			b_2m_ (mm^3^/(g·°C·bar))	4.5284 × 10^−4^	2.25 × 10^−5^	
			b_3m_ (mm^3^ /(g·°C·bar^2^))	1.11 × 10^−7^	1 × 10^−8^	
		(16)	a_1s_’ (mm^3^/g)	1119.039	1.293	99.43
		(18)	a_1s_’’ (mm^3^/g)	3.082	0.15	
		(20)	a_2s_ (mm^3^/(g·bar))	0.058	0.002	
		(21)	a_3s_ (mm^3^/(g·bar^2^))	6.272 × 10^−6^	7.13 × 10^−7^	
		(31)	b_1s_ (mm^3^/(g·°C))	0.508	0.018	
			b_2s_ (mm^3^/(g·°C·bar))	2.78594 × 10^−4^	3.005 × 10^−5^	
			b_3s_ (mm^3^/(g·°C·bar^2^))	6 × 10^−8^	1.1 × 10^−8^	
			c_1_ (mm^3^/g)	115.55	3.59	
			c_2_ (1/°C)	0.144	0.005	
			c_3_ (1/bar)	2.46612 × 10^−4^	2.7623 × 10^−5^	
	cal.2	(11)	a_1m_’ (mm^3^·bar/g)	21,997,152	27,796	99.79
		(12)	a_1m_’’ (mm^3^·bar/g)	−13,713	36,456	
		(14)	a_2m_ (bar)	18,064.71	90.388	
		(31)	b_1m_ (mm^3^·bar /(g·°C))	1001.603	105.288	
			b_2m_ (bar)	1032.106	4.03	
		(16)	a_1s_’ (mm^3^·bar/g)	28,469,060	340,822	99.19
		(17)	a_1s_’’ (mm^3^·bar/g)	83,976	4960	
		(19)	a_2s_ (bar)	25,644.56	327.387	
		(21)	b_1s_ (mm^3^·bar/(g·°C))	799.37	58.416	
		(31)	b_2s_ (bar)	1883.346	212.861	
			c_1_ (mm^3^/g)	120.666	4.016	
			c_2_ (1/°C)	0.13	0.005	
			c_3_ (1/bar)	3.81297 × 10^−4^	3.1849 × 10^−5^

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
