# Peer review of "Continuous Two-Domain Equations of State for the Description of the Pressure-Specific Volume-Temperature Behavior of Polymers"

_polymers, 2020, doi:10.3390/polym12020409_

Round 1
Reviewer 1 Report
This paper discussed the pvT behavior of amorphous ABS and semi-crystalline PP. The reviewer’s comments are showing below,
1.The reviewer suggests that the author should strengthen the novelty of this work in the introduction.
2.The authors should also confirm the glass transition behavior in Fig. 12 by DSC. In addition, the data of this work should be compared with the data measured by DSC. A comparative discussion should be made.
Author Response
Point 1: The reviewer suggests that the author should strengthen the novelty of this work in the introduction.
Response 1: Several sentences have been added to describe the novelty of this work in the introduction.
Point 2: The authors should also confirm the glass transition behavior in Fig. 12 by DSC. In addition, the data of this work should be compared with the data measured by DSC. A comparative discussion should be made.
Response 2: We added the values of the state transition temperature by DSC in Fig. 5. The prediction results by the pvT model are consistent with the DSC experimental results. The comparative discussion was added. Since the length of this manuscript is already too long, we are planning another manuscript which will mainly discuss the relationship between pvT measurement and DSC.
Reviewer 2 Report
Review Comments:
This paper presents a new continuous, two-domain pvT model that also includes the cooling rate effect. This represents a major step in providing a more accurate pvT model for the computer simulation of the injection molding process. Thus, this paper is suitable for being accepted and published in the journal of Polymers.
Due to the limitation of the experimental device used in this study, the highest cooling rate of 12.4 C/min was used. However, for injection molding, the cooling rate typically exceeds hundreds (if not thousands) of degrees C per min. Thus, it is recommended that the authors add a comment on how this continuous model can be applied to the real-world, ultra-high cooling rate situations?
Author Response
Point: Due to the limitation of the experimental device used in this study, the highest cooling rate of 12.4 C/min was used. However, for injection molding, the cooling rate typically exceeds hundreds (if not thousands) of degrees C per min. Thus, it is recommended that the authors add a comment on how this continuous model can be applied to the real-world, ultra-high cooling rate situations?
Response: DSC can be used to obtain the information about state transition of polymers, and high cooling rates can be realized. Therefore, we added the state transition temperature values including the data at high cooling rate (up to 50 °C/min) to confirm and validate the accuracy of the model. The results showed good agreement in Fig. 5. For the transition temperature at much higher cooling rates, new techniques such as flash DSC could be used. However, for the specific volume at higher cooling rates, it still needs more related data to validate the model, and then some related parameters of the model such as a1’, a1’’, a2 and a3 could be modified. We added a comment on it in section 3.5.